# Diversity-oriented synthesis of P-stereogenic and axially chiral monodentate biaryl phosphines enabled by C-P bond cleavage

Liangzhi Pang[1], Zhan Huang[1], Qilin Sun[1], Gen Li[1], Jiaojiao Liu[1], Baoli Li[1], Congyue Ma[1], Jiaxu Guo[1], Chuanzhi Yao[1], Jie Yu [1,2]✉ & Qiankun Li [1,2]✉

Chiral monodentate biaryl phosphines (MOPs) have attracted intense attention as chiral ligands over the past decades. However, the creation of structurally diverse chiral MOPs with both *P*- and axial chirality is still in high demand but challenging. Here, we show a distinct strategy for diversity-oriented synthesis of structurally diverse MOPs containing both *P*- and axial chirality enabled by enantioselective C-P bond cleavage. The key chiral Pd[II] intermediates, generated through the stereoselective oxidative addition of C-P bond, could be trapped by alkynes, $R_3Si$-Bpin, diboron esters or reduced by $H_2O/B_2pin_2$, leading to enantioenriched structurally diverse MOPs in excellent diastereo- and enantioselectivities. Based on the outstanding properties of the parent scaffolds, the *P*- and axially chiral monodentate biaryl phosphines serve as excellent catalysts in asymmetric [3 + 2] annulation of MBH carbonate affording the chiral functionalized bicyclic imide.

Chiral phosphines have found numerous applications in asymmetric catalysis used as ligands and catalysts, both on laboratory scale and in industrial processes, providing invaluable access to chiral molecules in a range of areas, including drugs and polymers as well as agrochemicals synthesis[1–6]. Thus the construction of chiral phosphines has been a field of intense research interest and development. During the infancy of the field, Knowles and co-workers successfully introduced the highly active DIPAMP ligand for rhodium catalyst in the first practical and highly enantioselective synthesis of L-DOPA[7,8]. Noyori and co-workers successfully introduced the BINAP in Ru-catalyzed asymmetric hydrogenation[9–11] casting a profound influence on asymmetric catalysis, thus leading to the demand for synthetic methods for accessing to chiral phosphine molecules. Among the chiral phosphines developed in the past decades, the exploited axial-chirality, C-stereogenic center, or planar chirality, are frequently used as the basic chiral elements constituting chiral phosphine ligands such as BINAP and DIOP. For an increasing number of asymmetric catalytic syntheses, combinations of the different chiral elements have proven

advantageous. For instance, TangPhos and DuanPhos featuring C-stereogenic centers and *P*-chiral phosphorus atoms, have found great applications in asymmetric catalysis[12].

On the other hand, chiral monodentate biaryl phosphines have attracted intense attention as chiral ligands over the past decades. For example, MOP[13] and KenPhos[14] with axial chirality, SegePhos[15] with point chirality, BI-DIME[16] with P-stereogenic center (Fig. 1a). However, phosphines containing both axial chirality and a P-stereogenic center have been less studied. In this regard, only a few types of such ligands were available due to the lack of general and efficient methods. Buchwald successfully prepared P-chirogenic binaphthyl-substituted monophosphines showing great application potential[17]. Cramer and coworkers disclosed the efficient and highly enantioselective C–H arylations of phosphine oxides with *o*-quinone diazides leading to the construction of monodentate chiral phosphorus[III] compounds with biaryl ligand backbones and chirality on *P*-atom (Fig. 1a)[18]. He and Liu reported the construction of P-stereogenic center and N–C axis enabled by asymmetric allylic substitution-isomerization[19]. Thus, the

[1]Department of Applied Chemistry, Anhui Agricultural University, 230036 Hefei, China. [2]School of Plant Protection, Anhui Province Engineering Laboratory for Green Pesticide Development and Application, and Anhui Province Key Laboratory of Crop Integrated Pest Management, Anhui Agricultural University, 230036 Hefei, China. ✉e-mail: jieyu@ustc.edu.cn; liqk@ahau.edu.cn

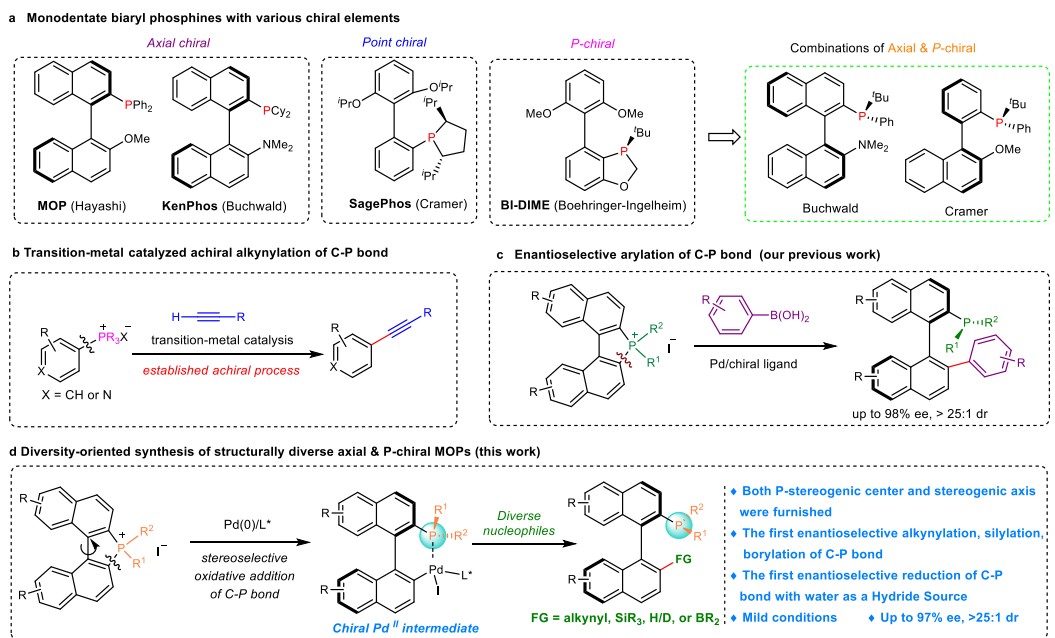

**Fig. 1 | Diversity-oriented synthesis of structurally diverse monodentate biaryl phosphines containing both *P*- and axial chirality. a** Monodentate biaryl phosphines with various chiral elements. **b** Transition-metal catalyzed achiral alkynylation of the C–P bond. **c** Enantioselective C–P bond arylation (our previous work). **d** Diversity-oriented synthesis of structurally diverse *P*- and axially chiral monodentate biaryl phosphines (this work).

development of efficient protocols, providing access to structurally diverse phosphines containing both P- and axial chirality is a highly desirable goal.

Due to the great interest and importance of axially chiral biaryl compounds[20–22] and monodentate biaryl phosphine ligands, we set our sights on developing a general and efficient synthetic method for the diversity-oriented synthesis of MOPs containing both *P*- and axial chirality through the stereoselective C–P bond cleavage reaction. The transition-metal-catalyzed C–P bond cleavage has emerged as a powerful tool for the formation of C–C and C–P bond[23–31]. Chang and coworkers have reported the achiral alkynylation of tetraarylphosphonium halides catalyzed by the transition metals[32]. Shen and coworkers reported the palladium-catalyzed Sonogashira coupling of heterocyclic phosphonium salts (Fig. 1b)[33]. Despite these advances, no examples of enantioselective C($sp^2$)–C($sp$) cross-coupling via asymmetric C–P bond cleavage have been reported so far. The principal challenge for the development of the desired enantioselective reaction was to achieve an efficient catalyst and chiral ligand. Another challenge was to realize both diastereo- and enantioselectivity. Very recently, we realized the palladium-catalyzed stereoselective C($sp^2$)–C($sp^2$) coupling of the C–P bond with aryl boronic acids for the enantioselective construction of chiral phosphines (Fig. 1c)[34]. We surmised whether the strategy could achieve the enantioconvergent C($sp^2$)–C($sp$) bond, C($sp^2$)–Si bond, C($sp^2$)–B bond, and C($sp^2$)–H bond to afford a new library of structurally diverse monodentate biaryl phosphines containing both *P*- and axial chirality with broad scope and functional group tolerance.

Here, we demonstrate the palladium-catalyzed formation of the C($sp^2$)–C($sp$) bond, C($sp^2$)–Si bond, C($sp^2$)-B bond, and C($sp^2$)–H bond via the stereoselective cleavage of the C–P bond, thus leading to the construction of structurally diverse monodentate biaryl phosphines containing a chiral axis and a *P*-chiral phosphorus atom (Fig. 1d). The key step of stereocontrol is realized via the stereoselective oxidative addition of C–P bond to the palladium/chiral ligand complex. The chiral Pd[II] intermediates could be then trapped by alkynes, $R_3Si$-Bpin, diboron esters, or reduced by $H_2O/B_2pin_2$, leading to enantioenriched structurally diverse monodentate biaryl phosphines in excellent diastereo- and enantioselectivities.

## Results

### Optimization of reaction conditions

Alkynes are among the most important class of compounds in all aspects of chemistry and are readily available starting materials for versatile transformations via the functionalization of the triple bond[35,36]. Though the achiral alkynylation of C–P bond has been achieved[32,33], no examples of C($sp^2$)–C($sp$) cross-coupling via asymmetric C–P bond cleavage have been reported. Thus we set our sights on developing the enantioselective alkynylation of *t*-butyl methyl biaryl phosphonium salt **1a** with terminal alkyne **2a**, leading to the alkynylated MOPs **3a** with an alkynyl group as a versatile handle for further synthetic elaborations to construct structurally diverse MOPs containing both *P*- and axial chirality.

The chiral phosphine ligands were first evaluated (Table 1, entries 1–6, for details, see Supplementary Information). To our delight, the reaction underwent smoothly to deliver the desired product **3a** in 86% yield and 91% ee using chiral phosphoramidite **L6** as the ligand (Table 1, entry 6). Further optimization of the additives indicated that CuBr was the best, providing **3a** in 90% yield and 91% ee (Table 1, entry 7). Other solvents were then surveyed but failed to improve the reactivity or enantioselectivity (Table 1, entries 9–12). The bases were also evaluated indicating that $K_2CO_3$ or $K_3PO_4$ gave better enantioselectivity but with lower yield (Table 1, entries 13 and 14). Further investigations revealed that 2.5 mol% catalyst loading with CuBr (1 equiv) was optimal providing the desired product **3a** in 83% yield and 93% ee at 45 °C for 36 h (Table 1, entry 18).

### Reaction scope of enantioselective alkynylation of C–P bond with terminal alkynes

With the optimal conditions in hand (Table 1, entry 18), we next set out to explore the substrate scope of this reaction by examining derivatives of biaryl phosphonium salts **1** and terminal alkynes **2**. Because of the product's sensitivity to air, they were more conveniently isolated as the corresponding air-stable phosphine sulfides or boranes by workup with $S_8$ or $BH_3$·$SMe_2$ respectively. Overall, all the tested substrates gave the expected MOPs in high yields with excellent atroposelectivities and enantioselectivities (Fig. 2). A wide range of terminal alkynes with different substituents or substitution patterns were evaluated under

**Table 1 | Optimization studies for enantioselective alkynylation of C–P bond with terminal alkynes**

| Entry | Ligand | Additive | Solvent | Base | 3a | | |
|---|---|---|---|---|---|---|---|
| | | | | | Yield (%)[a] | e.e. (%)[b] | dr |
| 1 | L1 | CuI | 2-Me-THF | Cs$_2$CO$_3$ | 65 | 51 | >25:1 |
| 2 | L2 | CuI | 2-Me-THF | Cs$_2$CO$_3$ | 18 | 54 | >25:1 |
| 3 | L3 | CuI | 2-Me-THF | Cs$_2$CO$_3$ | 93 | 8 | >25:1 |
| 4 | L4 | CuI | 2-Me-THF | Cs$_2$CO$_3$ | >99 | 86 | >25:1 |
| 5 | L5 | CuI | 2-Me-THF | Cs$_2$CO$_3$ | 42 | 82 | >25:1 |
| 6 | L6 | CuI | 2-Me-THF | Cs$_2$CO$_3$ | 86 | 91 | >25:1 |
| 7 | L6 | CuBr | 2-Me-THF | Cs$_2$CO$_3$ | 90 | 91 | >25:1 |
| 8 | L6 | CuCl | 2-Me-THF | Cs$_2$CO$_3$ | 47 | 90 | >25:1 |
| 9 | L6 | CuBr | TBME | Cs$_2$CO$_3$ | 19 | 95 | >25:1 |
| 10 | L6 | CuBr | DME | Cs$_2$CO$_3$ | 55 | 91 | >25:1 |
| 11 | L6 | CuBr | Toluene | Cs$_2$CO$_3$ | 16 | 89 | >25:1 |
| 12 | L6 | CuBr | CH$_3$CN | Cs$_2$CO$_3$ | Trace | N.D. | N.D. |
| 13 | L6 | CuBr | 2-Me-THF | K$_2$CO$_3$ | 78 | 93 | >25:1 |
| 14 | L6 | CuBr | 2-Me-THF | K$_3$PO$_4$ | 76 | 93 | >25:1 |
| 15 | L6 | CuBr | 2-Me-THF | DBU | 30 | 90 | >25:1 |
| 16 | L6 | CuBr | 2-Me-THF | Et$_3$N | 0 | N.D. | N.D. |
| 17[c] | L6 | CuBr | 2-Me-THF | Cs$_2$CO$_3$ | 98 | 92 | >25:1 |
| 18[c,d] | L6 | CuBr | 2-Me-THF | Cs$_2$CO$_3$ | >99 (83) | 93 | >25:1 |

The enantioselective alkynylation of C–P bond was performed by using **1a** (0.2 mmol), **2a** (0.4 mmol), [Pd(allyl)Cl]$_2$ (5 mol%), chiral phosphine (22 mol%), additive (0.75 equiv), and base (2.0 equiv) in solvent (2.0 mL) at 60 °C for 12 h.
N.D. not detected.

[a]The yield was determined by ¹H NMR using CH$_2$Br$_2$ as the internal standard. Isolated yields are reported in parentheses.

[b]ee values of the major isomers are shown and determined by chiral HPLC analysis. dr values were determined by ¹H NMR analysis of the crude reaction mixtures.

[c][Pd(allyl)Cl]$_2$ (2.5 mol%). **L6** (11 mol%) and the reaction was performed at 45 °C for 36 h.

[d]CuBr (1 equiv) was used.

**Fig. 2 | Substrate scope of enantioselective alkynylation of C−P bond.** Reaction conditions: **1** (0.20 mmol), **2** (2.0 equiv), [Pd(Allyl)Cl]₂ (2.5 mol%), **L6** (11 mol%), CuBr (1.0 equiv), Cs₂CO₃ (2.0 equiv), 2-Me-THF (2.0 mL) at 45 °C for 36 h, then S₈ (5 equiv) or BH₃·SMe₂ (2 equiv, 10 M in Me₂S), unless otherwise stated. Isolated yields were reported. ee values of the major isomers are shown and determined by chiral HPLC analysis. dr values were determined by ¹H NMR analysis of the crude reaction mixtures; ᵃ60 h; ᵇ**1** (0.05 M); ᶜthe reaction was conducted at 60 °C; ᵈwith bis(trimethylsilyl)acetylene as the coupling partner; ᵉ0.10 mmol scale.

the standard conditions (**3c**−**3s**). Aryl terminal alkynes that bear a substituent at the *ortho*, *para*, or *meta* positions were all well tolerated, and the corresponding products were furnished in high yields with high ee (**3c**−**3p**). The 2-ethynylnaphthalene was also tolerated, providing the desired product **3q** in 55% yield and 94% ee. Then 2-ethynylthiophene and 3-ethynylthiophene were evaluated, and the corresponding products were furnished in high yields with high ee (**3r**,

**3s**). In addition, the TMS-alkynyl product **3t** could also be afforded in 65% yield and 92% ee with bis(trimethylsilyl)acetylene as the coupling partner. Furthermore, several substituents were also well tolerated, such as fluoride (**3d**, **3p**), chloride (**3e**, **3o**), bromide (**3c**), OMe (**3g**, **3l**, **3n**), NH₂ (**3i**), OH (**3j**), and TMS (**3t**) functionality.

To further expand the generality of this strategy, a wide range of phosphonium salts **1** with different substituents were tested affording

**Fig. 3 | Substrate scope of enantioselective silylation of C−P bond.** Reaction conditions: **1** (0.20 mmol), R₃Si·Bpin **4** (2.0 equiv), [Pd(Allyl)Cl]₂ (2.5 mol%), **L6** (11 mol%), CuCl (0.75 equiv), Cs₂CO₃ (2.0 equiv), 2-Me-THF (2.0 mL) at 60 °C for 12 h, then S₈ (5 equiv) or BH₃·SMe₂ (2 equiv, 10 M in Me₂S), unless otherwise stated. Isolated yields were reported. ee values of the major isomers are shown and determined by chiral HPLC analysis. dr values were determined by ¹H NMR analysis of the crude reaction mixtures; ᵃthe reaction was conducted at 45 °C for 36 h; ᵇ0.10 mmol scale.

the corresponding products in high yields and excellent enantioselectivities of up to 97% ee (**3u**−**3ac**). The introduction of substituents at 7- and 7'-position on the aromatic ring of naphthalene was also applicable providing the desired product **3ae** in 63% yield with 95% ee. Furthermore, double-substituted phosphonium salts at 6- and 6'-postions were also investigated, and the corresponding products were observed in high yields with high ee values (**3af**−**3aj**). In addition, the hindered biphenyl phosphonium salts were also tolerated providing the desired products in moderate to good yields and high ee values (**3ak**−**3al**). Finally, the reaction was scaled up to the 2 mmol scale and the corresponding product **3a** was afforded an 82% yield with 92% ee. High diastereoselectivities (>25:1 dr) were obtained for most cases. The absolute configuration of the product was confirmed by single-crystal X-ray diffraction analysis (Figs. 2, 3c, CCDC 2227153).

### Reaction scope of enantioselective silylation of C−P bond
After the enantioselective alkynylation of the C−P bond was achieved, we became interested in developing an enantioselective silylation of the phosphonium salt **1** providing the corresponding MOPs **5** containing both P- and axially chirality. As important building blocks, organosilicon compounds have found numerous applications in organic synthesis, material science, and medicinal chemistry owing to their high stability and low toxicity[37,38]. Therefore, the development of

efficient protocols for the construction of C−Si bond is of great interest and desirable. In addition, the introduction of silicon groups into a chiral ligand or catalyst can tune the steric phenomenon thus enhancing the efficiency of the chiral induction. To our delight, the reaction of phosphonium salts **1** and silyborane **4** as the silicon source under similar conditions as the enantioselective alkynylation of C−P bond provides the desired products **5** in high yields and excellent enantioselectivities (Fig. 3). A wide range of phosphonium salts **1** with different substituents were tolerated affording the corresponding products in high yields and excellent enantioselectivities of up to 96% ee (**5a**−**5o**). The introduction of substituents on the aromatic ring of naphthalene was also applicable providing the desired products in high ees (**5p**−**5s**). In addition, the hindered biphenyl phosphonium salts were also tolerated providing the desired products in good yields and high ee values (**5t**−**5u**). The structure of the product was confirmed by single-crystal X-ray diffraction analysis, the stereogenic axis of **5o** was determined as R, and the chirality at the P-atom is the R configuration (Figs. 3, 5o, CCDC 2227154).

### Reaction scope of enantioselective borylation of C−P bond
Arylboronic esters play important roles in a variety of fields ranging from organic synthesis to drug discovery and material science[39–41]. Considering the generality of enantioselective alkynylation and

**Fig. 4 | Substrate scope of enantioselective borylation of C-P bond.** Reaction conditions: **1** (0.10 mmol), B$_2$neop$_2$ (2.0 equiv), [Pd(Allyl)Cl]$_2$ (5 mol%), **L5** (22 mol%), CuCl (0.75 equiv), Cs$_2$CO$_3$ (2.0 equiv), $p$-xylene (1.0 mL) for 12 h, then S$_8$ (5 equiv) or BH$_3$·SMe$_2$ (2 equiv, 10 M in Me$_2$S), unless otherwise stated. Isolated yields were reported. ee values of the major isomers are shown and determined by chiral HPLC analysis. dr values were determined by $^1$H NMR analysis of the crude reaction mixtures; [a] the reaction was conducted at 35 °C for 36 h; [b] 0.10 mmol scale.

silylation of C–P bond, we set our sights on developing the enantioselective borylation of biaryl phosphonium salts **1** leading to the *ortho*-borylated MOPs **6** with C–B bond as a versatile handle for further synthetic elaborations to construct structurally diverse MOPs containing both *P*- and axial chirality. To our delight, the reaction of phosphonium salts **1** and diboron esters B$_2$neop$_2$ as the boron source under the modified conditions provide the desired products **6** in moderate to high yields and excellent enantioselectivities (Fig. 4). In addition, the hindered biphenyl phosphonium salt was also tolerated providing the desired product **6g** in 41% yield and 96% ee. It should be noted that the enantioselective borylation of the C–P bond was much more sensitive to the space steric of the substitutes on the phosphorous atom. For example, the borylated product **6f** was only afforded in 38% yield but with high enantioselectivity of 86% ee.

## Reaction scope of enantioselective reduction of C–P bond with water as a Hydride Source

Diboron–H$_2$O has been recently used as a hydride source in Pd-catalyzed transfer hydrogenation of alkenes and alkynes by Stokes and coworkers[42]. Zhu and coworkers reported the elegant palladium-catalyzed enantioselective reductive Heck reactions via the reduction of C($sp^3$)–Pd bond with water as a hydride source assisted by diboron[43]. However, to the best of our knowledge, the enantioselective reduction of C($sp^2$)–Pd bond via the cleavage of C–P bond with water as a hydride source has never been achieved. Water is without doubt the most

**Fig. 5 | Substrate scope of enantioselective reduction of C–P bond.** Reaction conditions: **1** (0.20 mmol), B$_2$pin$_2$ (1.2 equiv), H$_2$O (4.0 equiv), [Pd(Allyl)Cl]$_2$ (5.0 mol%), **L6** (22 mol%), CuCl (0.75 equiv), Cs$_2$CO$_3$ (2.0 equiv), TBME (2.0 mL) at 60 °C for 12 h, then S$_8$ (5 equiv) or BH$_3$·SMe$_2$ (2 equiv, 10 M in Me$_2$S), unless otherwise stated. Isolated yields were reported. ee values of the major isomers are shown and determined by chiral HPLC analysis. dr values were determined by $^1$H NMR analysis of the crude reaction mixtures; [a] the reaction was conducted at 30 °C for 24 h; [b] the reaction was conducted at 45 °C for 24 h; [c] 0.10 mmol scale.

environmentally benign and cost-efficient hydride source. We became interested in developing an enantioselective reductive C−P bond cleavage of the phosphonium salt **1** thus providing the corresponding MOPs **7** containing both P- and axially chirality. To our delight, the reaction underwent smoothly to deliver the desired product **7a** in 91% yield and 91% ee under simple modified reaction conditions with TBME instead of 2-Me-THF as the solvent at 60 °C for 12 h. Then a wide range of phosphonium salts **1** with different substituents were tested affording the corresponding products in high yields and excellent enantioselectivities of up to 96% ee (Fig. 5). The introduction of substituents on the aromatic ring of naphthalene was also applicable providing the desired products in high ees (**7n**−**7s**). In addition, the hindered biphenyl phosphonium salts were also tolerated providing the desired products in good yields and high ee values (**7t**−**7u**). The stereogenic axis of **7h** was determined as R, and the chirality at the P-atom is the R configuration by X-ray diffraction analysis (CCDC 2227155).

## Synthetic utility and control experiments

To demonstrate the synthetic potential of this strategy, we readily converted the alkyne moiety in the enantio-enriched product **3a** to *E*-alkenyl group in **8** in one reduction step (Fig. 6a), which can offer an opportunity to access to *P*, alkene-bidentate ligand in the transition-metal catalyzed reactions. In addition, the formed C−B bond can be readily converted into C−O bond by oxidation of **6a** with NaBO$_3$·4H$_2$O providing the hydroxylation product **9** in 49% yield and 96% ee (Fig. 6b). Furthermore, the formed C−B bond can also be readily converted into C−H bond providing **7i** in 88% yield (Fig. 6c), thus the absolute configuration of boronated products **6** was confirmed as the same with that of products **7**. The stereogenic axis of **6** was determined as R, and the chirality at the P-atom is the R configuration.

For the enantioselective alkynylation of the C−P bond with terminal alkynes, a linear correlation between the ee values of the product **3a** and phosphoramidite **L6** was observed, suggesting that

monomeric PdL* (L* = phosphoramidite **L6**) might be involved in the enantio-determining steps (Fig. 6e). Treatment of **1a** using HCOONa as a hydride donor under the standard conditions afforded **7i** in 22% [1]H NMR yield with 67% ee (for details, see the Supplementary Information). **7i**-D bearing a D substituent was afforded with 90% D-incorporation using D$_2$O instead of H$_2$O under standard conditions (Fig. 6d), showing that H$_2$O is the hydride source of this reaction. In addition, **7i** was not observed when **6a** was treated with H$_2$O for 12 h under the standard reaction conditions, suggesting that protonolysis of **6a** is not responsible for the formation of **7i** (Fig. 6f). Furthermore, the reaction of **1a** with water in the absence of B$_2$pin$_2$ under standard conditions failed to afford compound **7i**, suggesting that B$_2$pin$_2$ plays an important role in the enantioselective reduction of C−P bond (Fig. 6g).

The chiral phosphines **L7**−**L11** were synthesized via the deprotection of the phosphine sulfides or boranes treated with Raney Ni or DABCO respectively (see Supplementary Information for details)[34]. With the rapidly constructed ligand library in hand, we next tested it in the asymmetric construction of functionalized bicyclic imides via [3 + 2] annulation of MBH carbonate (Table 2). For the reaction between the MBH carbonate **10** and *N*-methylmaleimide **11**, **MOP** was not effective and no desired product was observed (Table 2, entry 1). To our delight, with the phosphine **L11** as the catalyst, the functionalized bicyclic imides **12**[44] was obtained in 94% ee and 93% yield. This example proved the potential of the structurally diverse chiral monodentate biaryl phosphines in asymmetric catalytic reactions.

Based on the above control experiments, and the previous reports[32–34,43], a possible reaction mechanism was proposed to explain the reaction processes. As shown in Fig. 7, phosphonium salts ($S_a$)-**1a** and ($R_a$)-**1a** were in rapid equilibrium with each other via the rotation of the C−C single bond[45]. The preferentially oxidative addition of Pd$^0$ with substrate (*R*)-**1a** cleaved C−P bond *a* to form enantio-enriched biaryl intermediate **A**. the C-P bond *b* was in high steric hindrance due to the naphthyl and $^t$Bu groups, that's why the reaction shows high dr value.

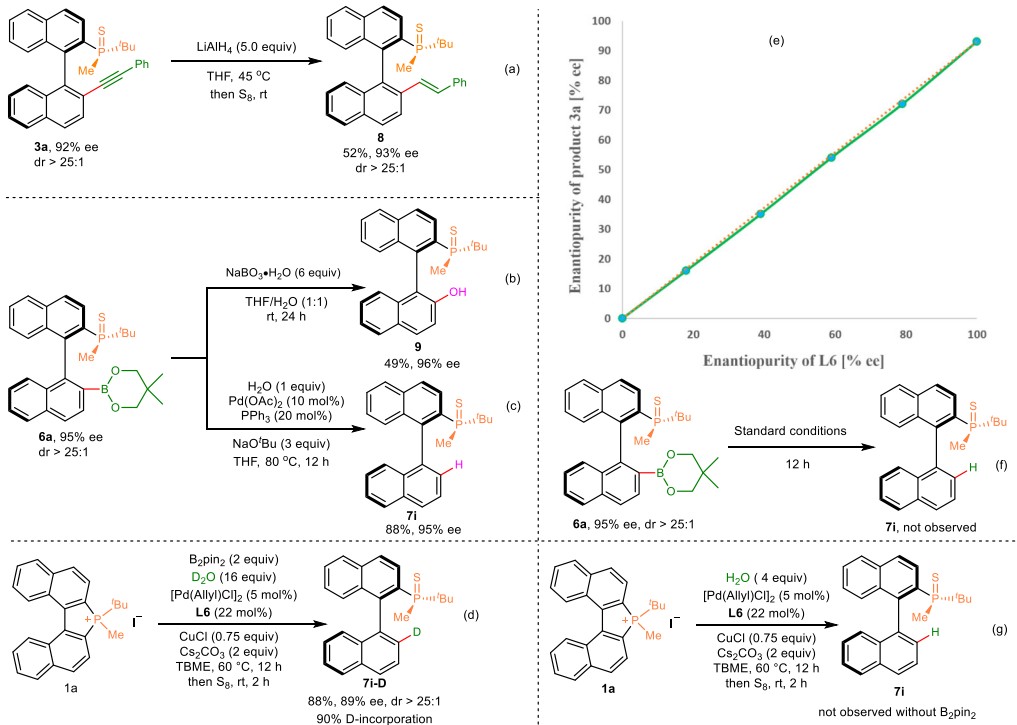

**Fig. 6 | Synthetic utility and control experiments. a** Reduction of the alkyne moiety to *E*-alkenyl group; **b** Oxidation of the C−B bond to C−O bond; **c** Converted the C−B bond to C−H bond; **d** Preparation of the D-substituent product; **e** The non-linear effect studies of the enantioselective alkynylation of C−P bond with terminal alkynes. **f** Control study of the protonation of C−B bond; **g** Control study of the reduction of C−P bond without B$_2$pin$_2$.

**Table 2 | Testing the developed ligand library in [3 + 2] annulation of MBH carbonate**

| Entry | Ligand | Yield (%) | e.e. (%) | dr |
|-------|--------|-----------|----------|-----|
| 1 | MOP | trace | N.D. | N.D. |
| 2 | L7 | trace | N.D. | N.D. |
| 3 | L8 | 92 | 67 | >25:1 |
| 4 | L9 | 40 | 84 | >25:1 |
| 5 | L10 | 98 | 83 | >25:1 |
| 6 | L11 | 93 | 94 | >25:1 |

Reaction conditions: **10** (0.1 mmol), **11** (0.12 mmol), 20 mol% L*, 1.0 mL toluene, room temperature, 22 h, under argon. Yields are those of isolated products.

**Fig. 7 | Proposed reaction mechanism for the enantioselective cross-coupling and reduction of C−P bond. Cycle I:** Phosphonium salts $(S_a)$-**1a** and $(R_a)$-**1a** are in rapid equilibrium with each other via the rotation of the C−C single bond. The oxidative addition of $Pd^0$ with substrate $(R)$-**1a** cleaves C−P bond $a$ to form enantio-enriched biaryl intermediate **A**. Subsequent transmetallation of intermediate **A** with the terminal alkynes, $R_3Si$-Bpin or diboron esters form the $Pd^{II}$ species **B**.

Reductive elimination of **B** furnishes the chiral phosphine product with concurrent regeneration of $Pd^0L^*$ catalyst. **Cycle II:** The reaction of $Pd^{II}$ intermediate **A** and water affords complex **C**. The transmetallation of complex **C** with the $B_2pin_2$ provides complex **D**. The reaction of complex **D** and water furnishes complex **E**. 1,4-D migration from **E** provides complex **F**. Reductive elimination of **F** delivers the desired product **7** with concurrent regeneration of $Pd^0L^*$ catalyst.

Subsequent transmetallation of intermediate **A** with the terminal alkynes, $R_3Si$-Bpin or diboron esters generated the $Pd^{II}$ species **B** (Fig. 7(I)). Reductive elimination of **B** furnished the chiral phosphine product, which would give the final product **3**, **5**, or **6** after being treated with $S_8$ or $BH_3$ with concurrent regeneration of $Pd^0L^*$ catalyst. In addition, as shown in Fig. 7(II), The reaction of $Pd^{II}$ intermediate **A** and water afforded complex **C**. The transmetallation of complex **C** with the $B_2pin_2$ provided complex **D**. The reaction of complex **D** and water would furnish complex **E**. 1,4-D migration from **E** provided complex **F**. Finally, reductive elimination of **F** would deliver the desired product **7** after treated with $S_8$ or $BH_3$.

## Discussion

In summary, we report an enantioselective and diastereoselective cleavage of the C−P bond for the synthesis of structurally diverse monodentate biaryl phosphines containing both *P*- and axial chirality which are expected to be useful chiral ligands or organocatalysts for enantioselective catalysis. These reactions feature broad substrate scope and mild conditions leading to the formation of $C(sp^2)$−$C(sp)$ bond, $C(sp^2)$−Si bond, $C(sp^2)$−B bond, and $C(sp^2)$−H bond. A series of structurally diverse monodentate biaryl phosphines are obtained in excellent yields with high diastereo-and enantioselectivities. The D-substituted monodentate biaryl phosphines can be easily synthesized using $D_2O$ as a D-donor with a high level of D-incorporation. The *P*- and axially chiral monodentate biaryl phosphines can serve as excellent catalysts in asymmetric [3 + 2] annulation of MBH carbonate affording the chiral functionalized bicyclic imide, which proves the potential of the structurally diverse chiral monodentate biaryl phosphines in asymmetric catalytic reactions.

## Methods
### Materials
Unless otherwise noted, materials were purchased from commercial suppliers and used without further purification. All the solvents were

treated according to general methods. Flash column chromatography was performed using 200−300 mesh silica gel. See Supplementary Methods for experimental details.

### Procedure for enantioselective alkynylation of C−P bond with terminal alkynes
To a 10 mL Schlenk tube containing anhydrous $Cs_2CO_3$ (0.4 mmol, 2 equiv) was added phosphonium salts **1** (0.2 mmol), CuBr (0.2 mmol, 1.0 equiv), [Pd(allyl)Cl]₂ (0.005 mmol, 2.5 mol%), chiral phosphine ligand **L6** (0.022 mmol, 11 mol%), 2-Me-THF (2 mL, 0.1 M), and terminal alkyne (0.4 mmol, 2 equiv) sequentially under nitrogen. The Schlenk tube was then sealed and stirred for 36 h at 45 °C. The reaction mixture was cooled to rt, $S_8$ (5 equiv, 1 mmol) or $BH_3$•$SMe_2$ (1 equiv, 10 M in $Me_2S$) was added and stirred for 2 h at rt. The reaction mixture was then filtered through a pad of celite eluenting with $CH_2Cl_2$/EtOAc (20 mL). The filtrate was concentrated, and the residue was purified by silica gel chromatography to afford the corresponding product.

### Procedure for enantioselective silylation of C-P bond
To a 10 mL Schlenk tube containing anhydrous $Cs_2CO_3$ (0.4 mmol, 2 equiv) was added phosphonium salts **1** (0.2 mmol), CuCl (0.15 mmol, 0.75 equiv), [Pd(allyl)Cl]₂ (0.005 mmol, 2.5 mol%), chiral phosphine ligand **L6** (0.022 mmol, 11 mol%), 2-Me-THF (2 mL, 0.1 M), and $R_3Si$-Bpin (0.4 mmol, 2 equiv) sequentially under nitrogen. The Schlenk tube was then sealed and stirred for 12 h at 60 °C. The reaction mixture was cooled to rt, $S_8$ (5 equiv, 1 mmol) or $BH_3$•$SMe_2$ (2 equiv, 10 M in $Me_2S$) was added and stirred for 2 h at rt. The reaction mixture was then filtered through a pad of celite eluenting with $CH_2Cl_2$/EtOAc (20 mL). The filtrate was concentrated, and the residue was purified by silica gel chromatography to afford the corresponding product.

### Procedure for enantioselective borylation of C−P bond
To a 10 mL Schlenk tube containing anhydrous $Cs_2CO_3$ (0.2 mmol, 2 equiv) was added phosphonium salts **1** (0.1 mmol), $B_2(OR)_2$ (0.2 mmol,

2 equiv), CuCl (0.075 mmol, 0.75 equiv), [Pd(allyl)Cl]$_2$ (0.005 mmol, 5 mol%), chiral phosphine ligand **L5** (0.022 mmol, 22 mol%), and *p*-xylene (1 mL, 0.1 M) sequentially under nitrogen. The Schlenk tube was then sealed and stirred for 36 h at 35 °C. The reaction mixture was cooled to rt, S$_8$ (5 equiv, 1 mmol) or BH$_3$•SMe$_2$ (2 equiv, 10 M in Me$_2$S) was added and stirred for 2 h at rt. The reaction mixture was then filtered through a pad of celite eluenting with CH$_2$Cl$_2$/EtOAc (20 mL). The filtrate was concentrated, and the residue was purified by preparative TLC.

### Procedure for enantioselective reduction of C–P bond with water as a Hydride Source

To a 10 mL Schlenk tube containing anhydrous Cs$_2$CO$_3$ (0.4 mmol, 2 equiv) was added phosphonium salts **1** (0.2 mmol), B$_2$pin$_2$ (0.24 mmol, 1.2 equiv), CuCl (0.15 mmol, 0.75 equiv), [Pd(allyl)Cl]$_2$ (0.01 mmol, 5 mol%), chiral phosphine ligand **L6** (0.044 mmol, 22 mol%), TBME (4 mL, 0.05 M), and H$_2$O (14 μL, 4 equiv) sequentially under nitrogen. The Schlenk tube was then sealed and stirred for 12 h at 60 °C. The reaction mixture was cooled to rt, S$_8$ (5 equiv, 1 mmol) or BH$_3$•SMe$_2$ (2 equiv, 10 M in Me$_2$S) was added and stirred for 2 h at r.t. The reaction mixture was then filtered through a pad of celite eluenting with CH$_2$Cl$_2$/EtOAc (20 mL). The filtrate was concentrated, and the residue was purified by silica gel chromatography to afford the corresponding product.

### Data availability

The authors declare that the data supporting the findings of this study are available within the article and its Supplementary Information file. For the experimental procedures, data of NMR and HPLC analysis, see Supplementary Methods and Charts in Supplementary Information file. The X-ray crystallographic coordinates for structures reported in this article have been deposited at the Cambridge Crystallographic Data Center (**3c:** CCDC 2227153, **5o:** CCDC 2227154, **7h:** CCDC 2227155). These data could be obtained free of charge from The Cambridge Crystallographic Data Center via www.ccdc.cam.ac.uk/data_request/cif.

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

## Acknowledgements
We are grateful for financial support from the National Natural Science Foundation of China (grant Nos. 92156022, 22001008, 22201009), Anhui Provincial Natural Science Funds (grant No. 1908085QB79), the University Synergy Innovation Program of Anhui Province (grant No. GXXT-2021-059), and Anhui Agricultural University.

## Author contributions
Q.L. and J.Y. conceived and directed the project. L.P. performed the reactions and control experiments. Z.H., Q.S., G.L., J.L., B.L., C.M., J.G., and C.Y. helped with the collection of new compounds and data analysis. Q.L. wrote the paper with input from all other authors. All authors discussed the results and commented on the manuscript.

## Competing interests
The authors declare no competing interests.
