## [Peer Review File · Nature Communications]

REVIEWER COMMENTS

Reviewer #1 (Remarks to the Author):

In this manuscript, the authors reported a distinct strategy for diversity-oriented synthesis of structurally diverse MOPs containing both P- and axial chirality enabled by enantioselective C-P bond cleavage. In this reaction, the key chiral Pd(II) intermediates, generated through the stereoselective oxidative addition of C-P bond, could be trapped by alkynes, R₃Si-Bpin, diboron esters or reduced by H₂O/B₂pin₂, leading to enantioenriched structurally diverse MOPs in excellent diastereo- and enantioselectivities. These experimental results further supported the C-P bond cleavage was suitable for the construction of axial chiral P-ligand. However, the chemistry reported in this manuscript is similarly to previous work that palladium-catalyzed stereoselective C(sp²)-C(sp²) coupling of C-P bond with aryl boronic acids for the enantioselective construction of chiral phosphines (Angew. Chem. Int. Ed. 2022, 61, e202211710). Although this work has many experimental data for the synthesis of structurally diverse MOPs containing both P- and axial chirality, the novelty reported in this manuscript is not enough for publication on Nature Communications. The manuscript might be suitable for the special journal in organic chemistry, such as Org. Chem. Front or J. Org. Chem..

Reviewer #2 (Remarks to the Author):

Chiral monodentate biaryl phosphines show valuable numerous applications in asymmetric catalysis used as ligands and catalysts. This paper described the palladium-catalyzed formation of C(sp²)-C(sp) bond, C(sp²)-Si bond, C(sp²)-B bond, and C(sp²)-H bond via the stereoselective cleavage of C-P bond for the construction of structurally diverse monodentate biaryl phosphines containing a chiral axis and a P-chiral phosphorus atom. The stereoselective oxidative addition of C-P bond to the palladium/chiral ligand complex formed the chiral PdII intermediates which were trapped by alkynes, R₃Si-BPin, diboron esters, or reduced by H₂O/B₂pin₂, leading to chiral monodentate biaryl phosphines in excellent diastereo- and enantioselectivities. The P- and axially chiral monodentate biaryl phosphines have been used as good chiral ligands in enantioselective [3+2] annulation of MBH. The nice original work seems to have been well conducted with sufficient details, and can be of utility to researchers interested in the organic chemistry and medicinal chemistry. Thus I recommend publication in this journal.

Some related article about the atroposelective synthesis of axially chiral biaryl compounds should be added ,such as Asymmetric synthesis of axially chiral compounds. Chin. J. Chem. 2021, 39, 1787; Synthesis of Atropisomers with Biaryl and Vinylaryl Chirality via Pd-Catalyzed Org. Lett., 2022.

Reviewer #3 (Remarks to the Author):

In this manuscript, Li and co-workers reported an enantioselective and diastereoselective cleavage of C-P bond. By using this strategy, the monodentate biaryl phosphines bearing both P- and axial chirality were synthesized efficiently. The work is based on the authors' previous work on the C-P cleavage chemistry. This paper herein further exhibited the broad substrate scope including alkynes, R₃Si-Bpin, diboron esters or protonation. Therefore, various MOPs could be generated easily, and there was no doubt about it: it could be served as the chiral ligands or Lewis base catalysts (The application was shown in Table 2). Overall, I would suggest the paper get in *Nat. Commun* after the minor revision.

- 1) The manuscript was focused on the binaphthalene ring. I would advise the author to add some examples on the hindered biphenyl ring for alkynylation, silylation, boronation and protonation. If the reactions work, the products are still to be axially molecules. In this case, the reaction would be more general.
- 2) In this context, the simultaneous construction of both P- and axial chirality were achieved. Thus, the recent paper (*Cell Rep. Phys. Sci.* 2021, 2, 100594) on the simultaneous construction of P-chiral center and stereogenic conformational axes should be cited in the paper.
- 3) The synthetic procedure of L7-L11 should be added in the manuscript or SI including their full characterization.

We would like to thank all the reviewers for your time and efforts in evaluating our manuscript (Manuscript ID: NCOMMS-23-04323-T). We have made all of the requested changes and revised the relevant part in the Manuscript and Supplementary information., which are listed point-by-point in this letter.

Reply to the comments of Reviewer #1

1. *Comment:* In this manuscript, the authors reported a distinct strategy for diversity-oriented synthesis of structurally diverse MOPs containing both P- and axial chirality enabled by enantioselective C-P bond cleavage. In this reaction, the key chiral Pd(II) intermediates, generated through the stereoselective oxidative addition of C-P bond, could be trapped by alkynes, R₃Si-Bpin, diboron esters or reduced by H₂O/B₂pin₂, leading to enantioenriched structurally diverse MOPs in excellent diastereo- and enantioselectivities. These experimental results further supported the C-P bond cleavage was suitable for the construction of axial chiral P-ligand. However, the chemistry reported in this manuscript is similarly to previous work that palladium-catalyzed stereoselective C(sp²)-C(sp²) coupling of C-P bond with aryl boronic acids for the enantioselective construction of chiral phosphines (*Angew. Chem. Int. Ed.* **2022**, *61*, e202211710). Although this work has many experimental data for the synthesis of structurally diverse MOPs containing both P- and axial chirality, the novelty reported in this manuscript is not enough for publication on Nature Communications. The manuscript might be suitable for the special journal in organic chemistry, such as Org. Chem. Front or J. Org. Chem.

Answer: We thank the reviewer's comments. However, the transition-metal-catalyzed cross-coupling of C-P bond could only be applied to construct C-C bond by now, the construct of C-Si, C-B bond via the cross-coupling of C-P bond has never been achieved. In this communication, we have firstly realized the enantioselective construction of C-Si, C-B bond via the cross-coupling of C-P bond leading to the formation of structurally diverse monodentate biaryl phosphines, with C-B bond or C-Si bond as versatile handles for further synthetic elaborations. In addition, the only example of the achiral reduction of C-P bond was reported by Yamamoto with high pressure (H₂, 100 atm) and high temperature (130 °C), and only one substrate was efficiently reduced. (Yamamoto, A. et al. *Chem. Lett.* **1995**, 1101-1102). In this communication, we have firstly realized the enantioselective reduction of C-P

bond with water as the environmentally benign and cost-efficient hydride source under mild conditions with high enantioselectivity.

Furthermore, the structurally diverse chiral monodentate biaryl phosphines have been proved very efficient in the [3+2] annulation of MBH carbonate with *N*-methylmaleimide to form the functionalized bicyclic imides in excellent yield and ee value. Thus the rapidly constructed ligand library will provide a large toolbox amenable for asymmetric catalysis.

In addition, the hindered biphenyl phosphonium salts were also tolerated providing the desired biphenyl phosphines in good yields and high ee values, which have not been reported in our previous report.

Reply to the comments of Reviewer #2

2. **Comment:** Chiral monodentate biaryl phosphines show valuable numerous applications in asymmetric catalysis used as ligands and catalysts. This paper described the palladium-catalyzed formation of C(sp²)-C(sp) bond, C(sp²)-Si bond, C(sp²)-B bond, and C(sp²)-H bond via the stereoselective cleavage of C-P bond for the construction of structurally diverse monodentate biaryl phosphines containing a chiral axis and a P-chiral phosphorus atom. The stereoselective oxidative addition of C-P bond to the palladium/chiral ligand complex formed the chiral Pd^{II} intermediates which were trapped by alkynes, R₃Si-BPin, diboron esters, or reduced by H₂O/B₂pin₂, leading to chiral monodentate biaryl phosphines in excellent diastereo- and enantioselectivities. The P- and axially chiral monodentate biaryl phosphines have been used as good chiral ligands in enantioselective [3+2] annulation of MBH. The nice original work seems to have been well conducted with sufficient details, and can be of utility to researchers interested in the organic chemistry and medicinal chemistry. Thus I recommend publication in this journal.

Answer: We thank the reviewer's comments.

3. **Comment:** Some related article about the atroposelective synthesis of axially chiral biaryl compounds should be added, such as Asymmetric synthesis of axially chiral compounds. Chin. J. Chem. 2021, 39, 1787; Synthesis of Atropisomers with Biaryl and Vinylaryl Chirality via Pd-Catalyzed Org. Lett., 2022. DOI:10.1021/acs.orglett.2c03760; Transition Metal-Catalyzed

Biaryl Atropisomer Synthesis, *Acc. Chem. Res.*, 2022, 55, 1620.

Answer: We have cited these papers about the atroposelective synthesis of axially chiral biaryl compounds as 20-22 in the References.

Reply to the comments of Reviewer #3

4. **Comment:** In this manuscript, Li and co-workers reported an enantioselective and diastereoselective cleavage of C-P bond. By using this strategy, the monodentate biaryl phosphines bearing both P- and axial chirality were synthesized efficiently. The work is based on the authors' previous work on the C-P cleavage chemistry. This paper herein further exhibited the broad substrate scope including alkynes, R_3Si -Bpin, diboron esters or protonation. Therefore, various MOPs could be generated easily, and there was no doubt about it: it could be served as the chiral ligands or Lewis base catalysts (The application was shown in Table 2). Overall, I would suggest the paper get in *Nat. Commun* after the minor revision.

Answer: We thank the reviewer's comments.

5. **Comment:** The manuscript was focused on the binaphthalene ring. I would advise the author to add some examples on the hindered biphenyl ring for alkynylation, silylation, boronation and protonation. If the reactions work, the products are still to be axially molecules. In this case, the reaction would be more general.

Answer: We thank the Referee for the helpful suggestions! We have added the following hindered biphenyl substrates. Generally, these products were afforded in moderate to good yields and high to excellent ees.

Scheme 1 Hindered biphenyl substrates.

6. Comment: In this context, the simultaneous construction of both P- and axial chirality were achieved. Thus, the recent paper (Cell Rep. Phys. Sci. 2021, 2, 100594) on the simultaneous construction of P-chiral center and stereogenic conformational axes should be cited in the paper.

Answer: We thank the Referee for the advices, we have cited this paper as 19 in the *References*.

7. Comment: The synthetic procedure of L7-L11 should be added in the manuscript or SI including their full characterization.

Answer: We have added the synthetic procedure of **L7-L11** and their characterization in the SI.

Deprotection of the phosphine sulphides and boranes

To a 25 mL Schlenk tube was added DABCO (67.2 mg, 0.60 mmol), **6b** (142.6 mg, 0.30 mmol), and toluene (2 mL) sequentially under nitrogen. The Schlenk tube was then sealed and stirred for 12 h at 50 °C. The reaction

mixture was cooled to rt and concentrated. The residue was purified by silica gel chromatography (petroleum ether /EtOAc 20:1) to afford the corresponding phosphine **L7** in 41% yield. Colorless oil, $R_f = 0.27$ (petroleum ether/ethyl acetate = 20:1), $^1\text{H NMR}$ (600 MHz, Chloroform-*d*) δ 7.97 (d, $J = 8.2$ Hz, 1H), 7.91 (d, $J = 8.2$ Hz, 1H), 7.89-7.81 (m, 3H), 7.77 (d, $J = 8.4$ Hz, 1H), 7.37 (t, $J = 7.2$ Hz, 2H), 7.19 (d, $J = 8.4$ Hz, 1H), 7.12 (q, $J = 8.0$ Hz, 2H), 7.07 (d, $J = 8.4$ Hz, 1H), 3.30 (d, $J = 10.8$ Hz, 2H), 3.22 (d, $J = 10.8$ Hz, 2H), 1.23 (d, $J = 5.1$ Hz, 3H), 0.77 (d, $J = 11.9$ Hz, 9H), 0.54 (s, 6H). $^{13}\text{C NMR}$ (151 MHz, Chloroform-*d*) δ 147.72 (d, $J = 33.2$ Hz), 144.40 (d, $J = 8.8$ Hz), 136.49 (d, $J = 20.1$ Hz), 134.31, 133.78 (d, $J = 7.1$ Hz), 133.30 (d, $J = 16.9$ Hz), 130.16, 128.85, 128.71, 128.69, 128.05, 128.03, 127.86, 127.39, 126.69, 126.20, 126.01, 125.98, 125.71, 124.94, 72.15, 31.39, 29.05 (d, $J = 14.3$ Hz), 28.29 (d, $J = 15.5$ Hz), 21.53, 8.41 (d, $J = 20.9$ Hz). $^{31}\text{P NMR}$ (243 MHz, Chloroform-*d*) δ -23.22. $[\alpha]_{\text{D}}^{25} = -15.3$ ($c = 0.575$, CH_2Cl_2). **HRMS (ESI)** calcd for: $\text{C}_{25}\text{H}_{27}\text{BO}_2\text{P}^+ [\text{M} - \text{C}_5\text{H}_8 + \text{H}]^+$ 401.1836; found: 401.1851.

To a 25 mL Schlenk tube was added DABCO (35.8 mg, 0.32 mmol), **5j** (83.2 mg, 0.16 mmol), and toluene (2 mL) sequentially under nitrogen. The Schlenk tube was then sealed and stirred for 15 h at 50 °C. The reaction mixture was cooled to rt and concentrated. The residue was purified by silica gel chromatography (petroleum ether / EtOAc 20:1) to afford the corresponding phosphine **L8** in 46% yield. Colorless oil, $R_f = 0.48$ (petroleum ether/ethyl acetate = 20:1), $^1\text{H NMR}$ (600 MHz, Chloroform-*d*) δ 7.96 (d, $J = 8.5$ Hz, 1H), 7.92-7.83 (m, 3H), 7.75-7.67 (m, 2H), 7.47-7.38 (m, 2H), 7.29 (d, $J = 6.9$ Hz, 2H), 7.24 (d, $J = 6.6$ Hz, 1H), 7.22-7.12 (m, 5H), 7.04 (d, $J = 8.5$ Hz, 1H), 3.14 (s, 3H), 3.05-2.94 (m, 2H), 1.40-1.22 (m, 4H), 1.14 (d, $J = 4.4$ Hz, 3H), -0.10 (s, 3H), -0.19 (s, 3H). $^{13}\text{C NMR}$ (151 MHz, Chloroform-*d*) δ 145.37 (d, $J = 31.8$ Hz), 144.94 (d, $J = 8.0$ Hz), 139.47, 137.55 (d, $J = 15.1$ Hz), 135.85 (d, $J = 2.8$ Hz), 134.25, 133.92 (d, $J = 6.5$ Hz), 133.65 (d, $J = 2.6$ Hz), 133.52 (d, $J = 15.7$ Hz), 131.93, 128.71, 128.26, 128.11, 127.81, 127.63, 127.60, 127.16, 126.81, 126.54, 126.45, 126.44, 126.34, 126.30, 125.83, 73.43 (d, $J = 11.4$ Hz), 58.37, 26.99 (d, $J = 13.6$ Hz), 25.74 (d, $J = 12.9$ Hz), 11.25 (d, $J = 16.4$ Hz), -1.26. $^{31}\text{P NMR}$ (243 MHz, Chloroform-*d*) δ -47.75. $[\alpha]_{\text{D}}^{25} = -17.6$ ($c = 0.375$, CH_2Cl_2). **HRMS (ESI)** calcd for: $\text{C}_{33}\text{H}_{36}\text{OPSi}^+ [\text{M} + \text{H}]^+$ 507.2268; found: 507.2279.

To a N₂-flushed Schlenk flask was loaded about 0.35 g of Raney Ni. The Raney Ni was washed with dried degassed CH₃CN (2 mL). To this flask was then added CH₃CN (2 mL) and **3g** (25.9 mg, 0.05 mmol). The resulting mixture was stirred under N₂ at rt for 14 h. The mixture was filtered through a short silica gel. The Raney Ni solid was washed with ether. The combined filtrate was concentrated under reduced pressure and the residue was purified by silica gel chromatography (petroleum ether /EtOAc 20:1) to afford product **L9** in 35% yield. White solid, R_f = 0.27 (petroleum ether/ethyl acetate = 20:1), ¹H NMR (600 MHz, Chloroform-*d*) δ 7.98 (d, *J* = 8.5 Hz, 1H), 7.95-7.84 (m, 4H), 7.74 (d, *J* = 8.4 Hz, 1H), 7.47-7.37 (m, 2H), 7.26-7.16 (m, 3H), 7.09 (d, *J* = 8.5 Hz, 1H), 6.82 (d, *J* = 8.1 Hz, 2H), 6.66 (d, *J* = 8.0 Hz, 2H), 3.70 (s, 3H), 1.25 (d, *J* = 4.5 Hz, 3H), 0.78 (d, *J* = 12.1 Hz, 9H). ¹³C NMR (151 MHz, Chloroform-*d*) δ 159.47, 145.28 (d, *J* = 33.3 Hz), 141.48 (d, *J* = 7.8 Hz), 137.10 (d, *J* = 20.8 Hz), 133.73, 133.34, 132.97 (d, *J* = 7.5 Hz), 132.91, 132.74, 128.90 (d, *J* = 3.2 Hz), 128.23, 128.15, 128.12, 127.88, 127.81, 127.45 (d, *J* = 2.1 Hz), 127.12, 126.50, 126.33, 126.09, 125.97, 122.88 (d, *J* = 5.3 Hz), 115.67, 113.88, 94.15, 89.17, 55.30, 29.16 (d, *J* = 13.8 Hz), 28.10 (d, *J* = 15.5 Hz), 8.36 (d, *J* = 20.6 Hz). ³¹P NMR (243 MHz, Chloroform-*d*) δ -22.06. [α]_D²⁵ = -74.8 (c = 0.086 CH₂Cl₂). HRMS (ESI) calcd for: C₃₄H₃₂OP⁺ [M + H]⁺ 487.2185; found: 487.2195.

To a N₂-flushed Schlenk flask was loaded about 0.9 g of Raney Ni. The Raney Ni was washed with dried degassed CH₃CN (2 mL). To this flask was then added CH₃CN (2 mL) and **7m** (51.7 mg, 0.13 mmol). The resulting mixture was stirred under N₂ at rt for 14 h. The mixture was filtered through a short silica gel. The Raney Ni solid was washed with ether. The combined filtrate was concentrated under reduced pressure and the residue was purified by silica gel chromatography (petroleum ether /EtOAc 20:1) to afford product **L10** in 97% yield. Colorless oil, R_f = 0.29 (petroleum ether/ethyl acetate = 20:1), ¹H NMR (600 MHz, Chloroform-*d*) δ 7.96 (d, *J* = 8.2 Hz, 2H), 7.92 (d, *J* = 8.1 Hz, 1H), 7.88 (d, *J* = 8.0 Hz, 1H), 7.75 (d, *J* = 8.1 Hz, 1H), 7.59 (t, *J* = 7.4 Hz, 1H), 7.48-7.36 (m, 3H), 7.25-7.18 (m, 2H), 7.18-7.10 (m, 2H), 3.14 (s, 3H), 3.11-2.99 (m, 2H), 1.63-1.50 (m, 1H), 1.45-1.30 (m, 3H), 1.22 (s, 3H). ¹³C NMR (151 MHz, Chloroform-*d*) δ 144.46 (d, *J* = 30.4 Hz), 137.85 (d,

$J = 8.0$ Hz), 133.52, 133.46, 133.26 (d, $J = 6.1$ Hz), 128.75, 128.72, 128.42, 128.18, 128.16, 127.88, 127.20 (d, $J = 2.2$ Hz), 126.58, 126.44, 126.37, 126.20, 126.20, 126.06, 125.85, 125.28, 73.33 (d, $J = 12.2$ Hz), 58.38, 25.86 (d, $J = 13.8$ Hz), 25.70 (d, $J = 12.7$ Hz), 12.90 (d, $J = 15.7$ Hz). ^{31}P NMR (243 MHz, Chloroform- d) δ -45.65. $[\alpha]_{\text{D}}^{25} = -36.7$ ($c = 0.462$, CH_2Cl_2). HRMS (ESI) calcd for: $\text{C}_{25}\text{H}_{26}\text{OP}^+$ $[\text{M} + \text{H}]^+$ 373.1716; found: 373.1719.

To a N_2 -flushed Schlenk flask was loaded about 2.4 g of Raney Ni. The Raney Ni was washed with dried degassed CH_3CN (5 mL). To this flask was then added CH_3CN (5 mL) and **7h** (134.8 mg, 0.34 mmol). The resulting mixture was stirred under N_2 at rt for 14 h. The mixture was filtered through a short silica gel. The Raney Ni solid was washed with ether. The combined filtrate was concentrated under reduced pressure and the residue was purified by silica gel chromatography (petroleum ether /EtOAc 20:1) to afford product **L11** in 78% yield. Colorless oil, $R_f = 0.78$ (petroleum ether/ethyl acetate = 20:1), ^1H NMR (600 MHz, Chloroform- d) δ 7.93 (d, $J = 8.2$ Hz, 2H), 7.90 (d, $J = 8.1$ Hz, 1H), 7.86 (d, $J = 8.0$ Hz, 1H), 7.78 (d, $J = 8.4$ Hz, 1H), 7.59 (t, $J = 7.0$ Hz, 1H), 7.45-7.36 (m, 3H), 7.25-7.09 (m, 4H), 1.59 (d, $J = 14.0$ Hz, 1H), 1.44 (d, $J = 13.9$ Hz, 1H), 1.26 (s, 3H), 0.51 (s, 9H). ^{13}C NMR (151 MHz, Chloroform- d) δ 143.59 (d, $J = 31.3$ Hz), 139.18 (d, $J = 14.0$ Hz), 137.94 (d, $J = 8.0$ Hz), 133.73, 133.48 (d, $J = 1.7$ Hz), 133.28, 133.26 (d, $J = 4.0$ Hz), 128.78 (d, $J = 3.7$ Hz), 128.36, 128.03, 128.01, 127.86, 127.26 (d, $J = 2.3$ Hz), 126.86, 126.75 (d, $J = 1.5$ Hz), 126.32, 126.30, 125.93, 125.76, 125.42, 47.49 (d, $J = 17.5$ Hz), 31.18 (d, $J = 14.1$ Hz), 30.71 (d, $J = 8.8$ Hz), 15.38 (d, $J = 15.6$ Hz). ^{31}P NMR (243 MHz, Chloroform- d) δ -54.30. $[\alpha]_{\text{D}}^{25} = -40.6$ ($c = 0.967$, CH_2Cl_2). HRMS (ESI) calcd for: $\text{C}_{26}\text{H}_{28}\text{P}^+$ $[\text{M} + \text{H}]^+$ 371.1923; found: 371.1932.

We thank all reviewers for their valuable comments that we believe substantially improved the quality of our manuscript.

With best wishes,

Qiankun Li, PhD

Department of Applied Chemistry
Anhui Agricultural University
Hefei, Anhui 230036, P. R. of China

REVIEWERS' COMMENTS

Reviewer #3 (Remarks to the Author):

Based on the revision of the paper, I would like to see the paper shown in Nat. Commun. at current version.